# Currently Applied Extraction Processes for Secondary Metabolites from *Lippia turbinata* and *Turnera diffusa* and Future Perspectives

Guillermo C. G. Martínez-Ávila [1], Pedro Aguilar-Zarate [2] and Romeo Rojas [1,*]

1   School of Agronomy, General Escobedo, Universidad Autonoma de Nuevo Leon, Monterrey 66050, Mexico; guillermo.martinezavl@uanl.edu.mx
2   Engineering Department, Tecnológico Nacional de México/Instituto Tecnológico de Ciudad Valles, San Luis Potosí 79010, Mexico; pedro.aguilar@tecvalles.mx
*   Correspondence: romeo.rojasmln@uanl.edu.mx; Tel.: +52-81-8329-4000 (ext. 3522)

**Abstract:** The poleo (*Lippia turbinata* Griseb.) and damiana (*Turnera diffusa* Wild) are two of the most valued species in the Mexican semidesert due to their medicinal uses. The conventional essential oil extraction process is hydrodistillation, and for the extraction of antioxidants, the use of organic solvents. However, these techniques are time-consuming and degrade thermolabile molecules, and the efficiency of the process is dependent on the affinity of the solvent for bioactive compounds. Likewise, they generate solvent residues such as methanol, hexane, petroleum ether, toluene, chloroform, etc. Therefore, in recent years, ecofriendly alternatives such as ohmic heating, microwaves, ultrasound, and supercritical fluids have been studied. These methodologies allow reducing the environmental impact and processing times, in addition to increasing yields at a lower cost. Currently, there is no up-to-date information that provides a description of the ecofriendly trends for the recovery process of essential oils and antioxidants from *Lippia turbinata* and *Turnera diffusa*. This review includes relevant information on the most recent advancements in these processes, including conditions and methodological foundation.

**Keywords:** *Lippia turbinata*; *Turnera diffusa*; essential oils; polyphenols; monoterpenes

## 1. Introduction

Damiana (*Turnera diffusa* Willd, family *Turneraceae*) is a deciduous shrub found in arid and semiarid regions of the West Indies, South America, Mexico, and the United States [1,2]. Its leaves are added as a condiment in food, and infusions are used for medicinal purposes, including nervous system stimulants, aphrodisiacs, and diuretics [1]. The main part is its leaves, which contain up to 1% essential oil formed of at least 20 bioactive compounds, including 1,8-cineole, α and β-pinene, p-cymene, thymol, calamene, alpha-copaene, tannins, flavonoids, damianin, beta-sitosterol, arbutin, glycosides, gonzalitosin, and tetraphyllin B, the last of which is suggested as a source of antioxidant properties [3]. Recent studies have shown that it has tyrosinase inhibitory properties [4], and offers testicular protection [1], prosexual effects in rats [5], recovery of sexual behavior [6], antiaromatase activity [7], and gastroprotective activity [8].

On the other hand, *Lippia turbinata* Griseb. (popularly known as "poleo") is an aromatic plant belonging to the *Verbenaceae* family, little documented by the literature. It is a native shrub from South America, commonly found in the northeastern region of Argentina [9] It is widely used in folk medicine to treat gastrointestinal disorders due to its antispasmodic properties, and in the food industry for its flavor [10,11]. Likewise, the harvesting season, geographical source, ripeness method, solvent, etc., influence the quantity and quality of essential oil yield, which contains 85–99% volatile and 1–15% nonvolatile components [12]. On the other hand, the volatile constituents are a mixture of terpenes, terpenoids, and other

aromatic and aliphatic constituents, all characterized by their low molecular weight [13]. In addition, some studies have shown that it has antifungal [11,14], virucidal [15], and insecticidal properties [16]. The major compounds identified by gas chromatography–mass spectrometry (GC–MS) in *L. turbinata* oil were piperitenone oxide (63.0%) and limonene (7.2%). Monoterpenes and their derivatives represented 78.6%, of which 71.4% were oxygenated monoterpenoids [17]. Given its rich and complex chemical composition, poleo represents an alternative to reducing the conventional additives in food matrices.

However, population growth and a changing lifestyle have given bioactive compounds (polyphenols and essential oils) great importance, from the food to the pharmaceutical sectors. Essential oils are aromatic compounds extracted from different parts of plants (leaves, seeds, roots, fruits, etc.), obtained by hydrodistillation (steam distillation) or solvent extraction (maceration). In addition, the essential oils of plants have valuable biological activities, such as antibacterial, antifungal, antiviral, and antioxidant properties, and represent a natural alternative to incorporate in foods, cosmetics, or drugs to improve, prevent, or treat several diseases [18]. The polyphenols are a superfamily of naturally occurring phytochemicals with antioxidant properties and health-regulating effects [19]. They have attracted much attention since their dietary consumption was associated with the prevention of some diseases such as severe acute respiratory syndrome coronavirus 2 (SARS-CoV-2) [20]. However, in many processes for extracting essential oils and phenolic compounds, organic solvents are used. Additionally, *Lippia turbinata* and *Turnera diffusa* are two species that are poorly explored for the recovery of bioactive compounds and essential oils by ecofriendly (no harm to the environment) technologies.

In recent years, ecofriendly technologies such as ohmic heating, microwave-assisted extraction, ultrasound, and supercritical fluids, among others, have been used for the recovery of bioactive compounds (essential oils and phenolic compounds) from different sources such as lavender [21], *Pulicaria undulata* [22], *Prangos ferulaceae* Lindle. [23], *Mentha piperita* [24], *Pinus pumila* [25], orange peels [26], *Perilla frutescens* (L.) Britt. [27], *Thymus vulgaris* L. [28] *Moringa oleifera* [29], apple seeds [30], *Thymus munbyanus* [31], *Diplotaenia cachrydifolia* [32], and others. These methodologies have proven to be more efficient, since they allow the reduction of time and the use of more suitable solvents, are less polluting, and increase yields [33]. Therefore, these methods show much promise as viable alternatives to recover high-quality essential oils. In their internal energy-generation system, the heating rate results in a short processing time and higher yields; however, the main disadvantage mentioned is the high cost of processing, but according to other studies, the use of emerging technologies increases the final cost by less than 3% [22,34,35]. This review focuses on the use of ecofriendly technologies for the recovery of bioactive compounds from *Lippia turbinata* and *Turnera diffusa*. Furthermore, because the information on the use of these technologies applied to the previously mentioned plants is very limited, it represents an opportunity to generate a starting point and promote their application in these highly important nontimber forest species.

## 2. Traditional Extraction Methods

Extraction of compounds, such as polyphenols and essential oils, from plants is an important field in which to obtain phytochemicals that have been obtained for decades by conventional extraction methods such as hydrodistillation, reflux, infusion, decoction, digestion, maceration, and percolation [33]. Each of these methods is briefly described below, emphasizing their foundation, advantages, and disadvantages, to give a general overview of the conventional extraction methods. In addition, the most used conditions (reported) for the extraction process of phenolic compounds and essential oils from *Turnera diffusa* and *Lippia turbinata* are summarized in Table 1.

### 2.1. Hydrodistillation

The main extraction method for the obtention of essential oils from several sources is hydrodistillation (water–steam). The process consists of immersing the sample in wa-

ter. The water is heated, generating vapors that carry the aromatic compounds. The vapor is then condensed to recover oily compounds [36]. However, these methods are time-consuming and involve large quantities of water, and induce the damage or loss of antioxidant activity in the bioactive compounds when using high temperatures [37,38]. This process allows the extraction of essential oil from both species using water as an extractant. The weight-to-volume ratio depends mainly on the capacity of the equipment used, ranging from 1:15 to 1:3 using a Clevenger-type apparatus for up to 3 h of processing. The maximum yields reach 10.9 μL of oil/g of plant (*Turnera diffusa*) and 10.2 μL of oil/g of plant (*Lippia turbinata*) (Table 1). These variations are mainly due to the source (plant), region, weather conditions, solvent, time, temperature, methodology, weight-to-volume ratio, etc. [39]. However, this species represents a viable alternative for the extraction of essential oils and phenolic compounds, as well as the identification of the compounds and their separation for their use in the food industries to the pharmaceutical industry.

## 2.2. Infusion

A common practice in rural communities is the preparation of infusions for medicinal purposes. The use of the infusion can be via ingestion or dermic use, depending on the type of plant and the solvent used (EtOH, MetOH, or water). The preparation consists of very hot or boiling water and vegetal material (leaves, lowers, fruits, seeds, or some barks of plants) to dissolve the soluble fraction of the components. Time, temperature, solvent, *w:v* rate, and stirring are variable, depending on the plant, type of compounds, particle size, etc. [40]. Only *Turnera diffusa* has been used in weight-to-volume ratio, from 1:10 to 1:22.5, or percolation overnight, with yields from 0.41 to 33.85 mg/g of plant. In addition, the TPP present an increasing sexual activity in male rats and antioxidant activity (Table 1). In practice, this is a method that has several disadvantages, since the mass:volume ratio, contact time, temperature, etc. are not controlled (unless it is an industrial-level process). All this mainly affects the quantity and quality of the extracted compounds. In addition, it does not require any type of specialized equipment in order to be carried out. However, for the recovery of phenolic compounds, chromatography columns are used with resins such as XAD-16 for the compound's purification. Likewise, this increases processing time, without forgetting the possible residuality of solvents (EtOH, MetOH, etc.).

## 2.3. Reflux

The reflux system is another method used for the recovery of bioactive compounds from various sources. It differs in configuration from the hydrodistillation extraction process due to the introduction of a reflux condenser, which functions in both the cooling and condensation of bioactive compounds [41]. It allows the stability of the compounds to be maintained, since it avoids the overheating of the sample and an eventual variation in their quality. However, large amounts of solvent are used, and recovery times are very long (up to 8 h) [38,42]. *Turnera diffusa* has been used with EtOH at a weight-to-volume ratio of 1:4. The yields were from 96.4 to 590 mg/g of plant, with 9.64 to 236.27 mg/g GAE and up to 91.96% DPPH inhibition (Table 1). It is important to mention that for *Lippia turbinata*, this has not been reported previously. This method has another disadvantage; that is, the use of organic solvents such as ethanol, methanol, ethyl ether, petroleum ether, and others, which cause environmental damage and increase operating costs [43]. However, these methodologies are still used due to the high availability of the equipment.

## 2.4. Soxhlet Extractor

The Soxhlet extractor was invented by Franz von Soxhlet in 1879. Today, Soxhlet extraction represents one of the most classical methodologies for compound extraction [44], and is described as the universal chemical extraction process [45]. By itself, it is an optimized extraction process, but the literature offers a high number of examples using different Soxhlet extraction conditions. This method is based on the separation of a specific fraction from several food or plant materials with the use of a polar solvent, depending

on the solubility characteristic of the target compounds and the physicochemical nature of source, which can determine the surface contact and diffusivity of the solvent into the samples. Commonly, for *Turnera diffusa*, EtOH 50% with a weight-to-volume ratio of 1:66.6 is used. The yields are 409 mg/g of plants with 161.62 mg GAE/dw, with higher antioxidant activity (Table 1). However, it requires long extraction times and large quantities of solvents such as methanol [46,47], ethanol [48,49], n-hexane [46,50], petroleum ether [51], toluene, chloroform [52], and others [53]. This makes the process expensive and unsuitable, although the equipment is very easy to obtain and operate.

### 2.5. Maceration

Maceration involves the storage of crushed plant material with a solvent for minutes, hours, or days. Heat can be applied to induce cell destruction, but this can degrade bioactive compounds. Since it weakens the cell walls and cytoplasmic membranes, which are resistant to mass transfer, a greater quantity of compounds can be extracted [54,55]. For the recovery of essential oils, the weight/volume ratio is not available. However, it was reported for the recovery of polyphenols (1:5 *w/v*). The yields were not reported, but the TPP had gastroprotective and antispasmodic activity (Table 1). This is a very simple process, since it is not necessary to use equipment; however, it is time-consuming, and it is necessary to use organic solvents to recover a greater quantity of compounds. For this reason, it is one of the most used in traditional medicine. According to the most recent literature, it has only been used to recover TPP from *Lippia turbinate*.

**Table 1.** Polyphenols and essential oils from *Turnera diffusa* and *Lippia turbinata* extracted for conventional methods.

| Extract | Method | Solvent | Conditions | Yield | Main Results | Ref. |
|---|---|---|---|---|---|---|
| | | | | *Turnera diffusa* | | |
| Oil | Distillation | $H_2O$ | 1:15 *w/v*, 1.5 h in a CTA | 10.9 ± 6.0 µL/g of plant | 1,8-cineole (17.20 ± 8.56%), 10-epi γ eudesmol (4.54 ± 0.49%), oplopenone (3.63 ± 0.37%) and aristolene (3.47 ±1.17%) | [56] |
| | | | 1:3 *w/v* ratio, 1.5 L/leaves and stems in a CTA | 0.158 mL/g of plant | 1,8-cineole; Bicyclo[3.1.1]heptan-3-ol, 6,6-dimethyl-2-methylene; Bicyclo[3.1.1]hept-2-en-6-one, 2,7,7-trimethyl and 1,4-Methanocycloocta[d]pyridazine, 1,4,4a,5,6,9,10,10a-octahydro-11,11-dimethyl-, (1à,4à,4aà,10aà) | [57] |
| | | | 500 g leaves/1 h in a CTA | Approximately 0.002 mL/g of plant | 1,8-cineole (7.1%) and thymol (5.1%) | [58] |
| TPP | Infusion | MetOH | 1:10 *w/v* ratio,28 °C/24 h/ 250 rpm | 0.410 ± 0.0039 mg/g of plant | 0.0080 mg/g of TFC. SRSA and ion chelation were higher in MetOH extract, and the phagocytosis activity increased in those leukocytes stimulated | [59] |
| | | $H_2O$ | 1.22.5 *w/v* ratio,60 °C/1 h stirred | 33.85 mg/g of plant | 72.32% of ABTS•+ inhibition; FRAP 21.33 mg GA/g | [57] |
| | | MetOH: $H_2O$ | Percolation overnight | NR | Sexually potent and sexually sluggish/impotent male rats were treated orally with different amounts. | [60] |
| | Reflux | EtOH | 2 g/80 °C/3 h | 590 ± 16.4 mg/g of plant | 236.27 ± 0.36 mg GAE/dw of TPC and 377.21 ± 0.08 mg Trolox/dw | [39] |
| | | EtOH 70% | 1:4 *w/v* ratio, 60 °C/2 h | 96.4 ± 31.1 mg/g of plant | 9.64 ± 3.11 mg GAE/g; 76.03% to 91.96% DPPH inhibition; 65% in the LOI and 50% in ABTS•+. Quercetin was identified. | [61] |
| | Soxhlet | EtOH 50% | 1.66.6 *w/v* | 409 ± 12.1 mg/g of plant | 161.62 ± 0.12 mg GAE/dw of TPC and 186.62 ± 0.007 mg Trolox/dw | [39] |
| | | | | *Lippia turbinata* | | |
| Oil | Distillation | $H_2O$ | 100 g/3 h in a CTA | 10.2 ± 1.1 µL/g of plant | Limonene was the main component. TPC 14.03 ± 0.12 mg GAE/100 g fresh vegetal material | [11] |
| | Distillation | $H_2O$ | 100 g/3 h in a CTA | 10.2 µL/g of plant | Limonene (48.83%), β-caryophyllene epoxide (18.06%), and piperitenone (7.67%) | [14] |
| | Distillation | $H_2O$ | 2 h in a CTA | NR | α-Thujone (48.3%), Carvone (17.4%), β-Caryophyllene (10.0%), Limonene (3.5%), α-Copaene (3.1%) | [16] |
| | Distillation | $H_2O$ | 2 h in a CTA | NR | (4R)(+)-Pulegone (3.56%) | [62] |
| | Distillation | $H_2O$ | Using a CTA | NR | Limonene (60.8%), Bornyl acetate (8.2%) and Eucarvone (5.8%) | [63] |
| NR | NR | MeOH-$CH_2Cl_2$ (1:1) | NR | NR | Four novel triterpenoids 3β,25-epoxy-3α,21α-dihydroxy-22β-(3-methylbut-2-en-1-oyloxy)olean-12-ene-28-oic acid (1); 3β,25-epoxy-3α,21α-dihydroxy-22β-angeloyloxyolean-12-ene-28-oic acid (2); 3β,25-epoxy-3α,21α-dihydroxy-22β-tigloyloxyolean-12-ene-28-oic acid (3); and 3α,25-epoxy-3α-hydroxy-22β-(2-methylbutan-1-oyloxy)olean-12-ene-28-oic acid (4) | [64] |
| TPP | Maceration | EtOH 50% | 1:5 *w/v*, 24 h Rotavapor 70 °C | NR | Gastroprotective and antispasmodic activity was evaluated. | [55] |

**TPP**: total polyphenols; **CTA**: Clevenger-type apparatus; **TFC**: total flavonoids content; **SRSA**: superoxide radical scavenging; **TPC**: total phenolic content; **MetOH**: methanol; *w/v*: weight/volume; dw: dry weight; **GAE**: gallic acid equivalents; **NR**: not reported; ABTS+: (2,2′-azino-bis(3-ethylbenzothiazoline-6-sulfonic acid); **DPPH**: 2,2-diphenyl-1-picrylhydrazyl; **FRAP**: ferric-reducing antioxidant power; **LOI**: lipid oxidation inhibition.

*2.6. The Most Important Compounds Identified and Their Properties*

Eucalyptol (1,8-cineole) was the main compound identified in *Turnera diffusa* essential oils, in concentrations up to 17.20%. This compound is a cyclic-ether monoterpene also found in several plants such as eucalyptus, rosemary, sage, bay, cinnamon, and tea [65]. In addition, this compound has been reported for its pharmacological properties, including analgesic [66], insecticidal [67], sedative [68], antioxidant [69], anti-inflammatory [70,71], bactericidal [72], fungicidal [73], hypolipidemic [74], and anticancer activities [75]. Another very important compound in various areas is thymol, which is a volatile compound that widely exists in the essential oils of thyme and oregano plants. It is classified as a generally recognized as safe (GRAS) compound by the US Food and Drug Administration. It possesses notable antimicrobial activities against bacteria and fungi [76], and is used in edible films [77] and other areas. In phenolic compounds, the main identified compound was quercetin (Figure 1). It is a flavonoid detected in fruits, vegetables [78], and more than 20 plants that have been traditionally suggested for their analgesic, antispasmodic, and antidiabetic properties, and for treatment of iron deficiency and many other disorders [79]. In addition, anti-inflammatory, antihypertensive, vasodilator, antiobesity, antihypercholesterolemic and antiatherosclerotic functions of this substance have been reported [80,81].

1,8-cineole

10-epi γ eudesmol

Oplopenone

Aristolene

Bicyclo[3.1.1]heptan-3-ol, 6,6-dimethyl-2-methylene

Bicyclo[3.1.1]hept-2-en-6-one, 2,7,7-trimethyl

1,4-Methanocycloocta[d]pyridazine, 1,4,4a,5,6,9,10,10a-octahydro-11,11-dimethyl-, (1à,4à,4aà,10aà)

Thymol

Quercetin

**Figure 1.** The main compounds identified in essential oils and polyphenols from Turnera diffusa extracted by conventional methods.

Limonene was the main compound identified in *Lippia turbinata* essential oils, in concentrations up to 48.83%. This compound is a terpene (Figure 2) that is composed of combinations of several five-carbon base ($C_5$) units called isoprene. The main terpenes are the monoterpenes ($C_{10}$) and sesquiterpenes ($C_{15}$). Terpenoids are terpenes containing oxygen. Monoterpenes, formed from the coupling of two isoprene units, are the

most representative molecules, constituting 90% of the essential oils [12], and have a variety of applications, including pharmaceutical, nutraceutical, agriculture, and flavor and fragrance [82]. Specifically, limonene is a typical monoterpene existing in more than 300 plants (e.g., lemon waste [83], orange peels [84], and others such as caraway seed [85] and *Agastache mexicana* [86]). Some microorganisms are used to obtain limonene through biotransformation, such as *Yarrowia lipolytica* [87,88], *Saccharomyces cerevisiae* [89,90], *Colletotrichum nymphaea* [91], *Synechococcus elongatus* [82], *E. coli*, and *Bacillus megaterium* [55]. Additionally, it is one of the most important exogenous biomarkers denoting a deficient liver metabolism, and is accumulated due to the liver's incapacity to convert it into carveol metabolites or perillyl metabolites by CYP2C enzymes [92–95].

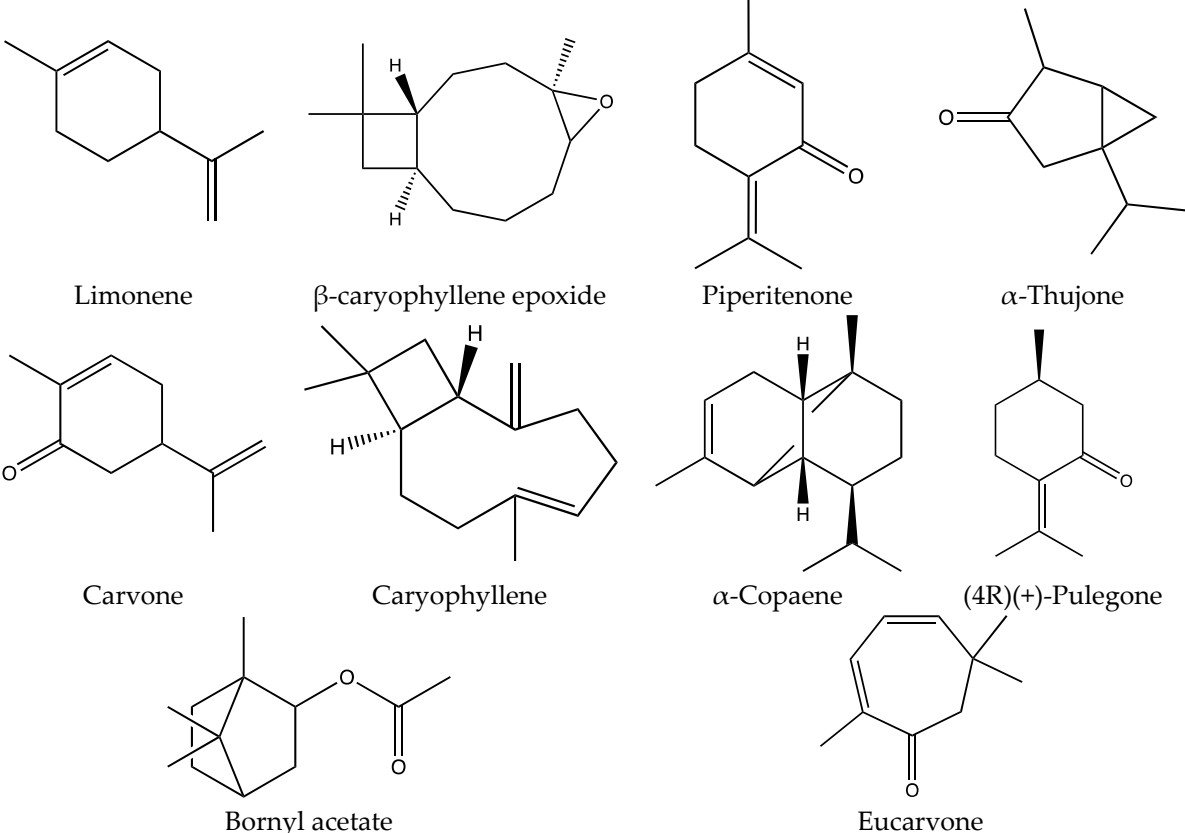

**Figure 2.** The main compounds identified in essential oils and polyphenols from Lippia turbinata extracted for conventional methods.

Limonene's derivatives, such as perillyl alcohol, menthol, carveol, and α-terpineol, are widely applied in food, pharmaceuticals, cosmetics, biomaterials, and biofuels [96–98]. α-Thujone (a monoterpenoid ketone) is considered a toxic compound [99] in humans and the cause of a syndrome called "absinthism", which was described in the 19th century after chronic abuse of the thujone-containing spirit [100]. However, in animals, thujone inhibits the γ-aminobutyric acid-A (GABA_A) receptor, causing excitation and convulsion in a dose-dependent manner [101]. The sesquiterpenes β-caryophyllene epoxide and β-caryophyllene are common flavoring and fragrance materials that are woody/spicy. They are found naturally in a variety of foods and spices (cinnamon, citrus fruits, clove, curry, sage, and thyme), and are commonly used in foods in the United States, Europe, and several regions around the world [102]. In Mexico, the extraction of essential oil from *Lippia turbinata* is not carried out, although the plant grows in the semidesert area in the northeast of the country and is used in traditional medicine. It is also worth mentioning that there are no current exploitation permits approved by CONAFOR (National Forestry

Commission). In 2001, four new compounds were identified in *Lippia turbinata*, in order to find compounds for pharmaceutical purposes from desert and semi-desert plants of Latin America (Figure S1). Through an extraction with MeOH-CH2Cl2, M. tuberculosis was completely inhibited at 100 μg/mL (this extract contained the new four compounds), showing the antimycobacterial activity for *Lippia* species for the first time [64]. Therefore, this is an opportunity area for the study and application of this species that is little valued in Mexico.

## 3. Ecofriendly Extraction Methods Used for *Lippia turbinata* and *Turnera diffusa*

Ecofriendly methods have advantages such as short extraction times, reduction of the use of organic solvents and environmental impact, and increased yields. These results are dependent on the sample. Some of the most used technologies to extract total polyphenols and essential oils are microwave, ultrasound, ohmic heating, supercritical fluids, and biotechnological extraction.

### 3.1. Microwave Assisted Extraction

Microwaves are electromagnetic energy with frequencies from 300 MHz to 300 GHz. They are transmitted in the form of waves and penetrate biomaterials, interacting with free water molecules to generate heat, causing focused heating. Therefore, the temperature increases rapidly to near or above the boiling point of water, generating a rapid expansion that leads to the rupture of cell walls [103]. Lipids have a low specific heat; therefore, they are susceptible to this radiation. This gives rise to permanent pores in the plant material and allows increasing yields [104]. For the extraction of essential oils, a microwave-assisted hydrodistiller is used. The plant material and water are heated using a domestic microwave oven modified with a lateral orifice to connect the flash and the condenser operating at full power (2.45 GHz, 800 W/30 min), and then decanted and dried with anhydrous sodium sulfate [105].

The use of emerging technologies for obtaining essential oils and polyphenols from *Turnera diffusa* and *Lippia turbinata* is very limited (Table 2). Nowadays, there are two reports related to the extraction of *Turnera diffusa* essential oil (7 μL/g) and polyphenols (606 mg/g) by ecofriendly methods [39,103]. The essential oil yields are very similar to what has been reported by conventional methods (see Tables 1 and 2). However, it is important to mention that the use of microwaves brings many advantages in that it is an ecofriendly method, it eliminates the use of organic solvents, and it reduces processing times. Likewise, it maintains the concentration of the polyphenolic compounds. Due to this, a window of possibilities for the use of emerging technologies for the extraction of oils and polyphenols is opened for these little-valued species. It is worth mentioning that in Mexico, there are no current exploitation permits for these two non-timber forests species, but they are used in medicine and traditional foods.

**Table 2.** Polyphenols and essential oils from damiana (*Turnera diffusa*) extracted by ecofriendly methods.

| Extract | Method | Solvent | Conditions | Yield | Main Results | Ref. |
|---------|--------|---------|------------|-------|--------------|------|
| Oil | MAE | $H_2O$ | 2.45 GHz, 800 W/30 min | 7 μL/g of plant | Drima-7,9(11)-diene (22.9%), β-viridiflorene (6.6%), α-silinene (5.9%), valencene (5.5%) | [105] |
| TPP | MAE | $H_2O$ | 2.75 g/300 W/220 rpm/ 50 °C/15 min | 606 ± 15.8 mg/g of plant | 239.52 ± 0.31 mg GAE/dw of TPC and 116.24 ± 0.08 mg Trolox/dw | [39] |
| | UAE | EtOH 50% | 2 g/40 °C | 516 ± 16.7 mg/g of plant | 203.96 ± 0.35 mg GAE/dw of TPC and 201.94 ± 0.07 mg Trolox/dw | [39] |

**TPP**: total polyphenols; **TPC**: total phenolic content; **dw**: dry weight; **GAE**: gallic acid equivalents; **EtOH**: ethanol; **MAE**: microwave-assisted extraction; **UAE**: ultrasound-assisted extraction.

### 3.2. Ultrasound-Assisted Extraction

UAE is used in processes for the extraction of plant compounds, reducing times, solvents and yields through a simpler process than conventional methodologies. The technique is based on cavitation, which is responsible for the implosion of microbubbles, causing the cell walls of plant tissue to rupture. This damage increases turbulence and penetration of the solvent into the plant matrix and causes the release of intracellular content, and increases the rate of mass transfer of the solvent to the internal area of the matrix and of the solvent soluble components. In addition, cavitation on the surface of the cell walls causes the disturbance of their structure, caused by solvent microjets, which also contributes to the increased mass transfer [106]. For the extraction of polyphenols from *Turnera diffusa*, the sample and solvent (EtOH 50%) are placed into a glass container and inserted into the ultrasonic device, then the temperature is set to 40 °C, in order to avoid the degradation of the bioactive compounds. The sample is allowed to cool at room temperature, then the solvent is evaporated using a rotary evaporator and the extracts are dried until reaching a constant weight [39]. By applying ultrasound-assisted extraction with *Turnera diffusa*, it is possible to obtain 516 mg/g of polyphenols (Table 2) [39]. In this way, the use of alternative methods has assumed a larger importance versus the disadvantages of conventional methods, due to avoiding compound degradation, and reducing the processing time and the use of organic solvents.

The main identified compounds in essential oils from *Turnera diffusa* (Figure 3) were different from those extracted by conventional methods (Drima-7,9(11)-diene, β-viridiflorene, α-silinene, valencene). The information on Drima-7,9(11)-diene and α-silinene is very limited. However, based on traditional medicinal knowledge, β-viridiflorene was scrutinized virtually against four structural protein targets of SARS-CoV-2 viz. 3CL$^{pro}$, ACE-2, spike glycoprotein, and RdRp [107]. Valencene, the chemical responsible for the fresh odor of oranges, is a sesquiterpene that is easily obtained from citrus fruits and is an inexpensive raw material that can be oxidized into high-value products such as nootkatone (which is another important sesquiterpenoid aroma component of the grapefruit) [108,109]. It is widely used in the flavor and fragrance industries. These molecules, based on interaction energies, conventional hydrogen bonding numbers, and other noncovalent interactions, in comparison with the known SARS-CoV-2 protease inhibitor (lopinavir and RdRp inhibitor), can be a phenomenal inhibitor of both protease and polymerase, as it strongly interacts with their active sites and can exhibit a remarkably high binding affinity. Furthermore, in silico drug-likeness and ADMET prediction analyses clearly evidenced the usability of this bioactive compound to develop as a drug against COVID-19 [107]. Finally, the efficiency of an extraction method is designated by high-performance total phenolic compound values in terms of weight of the dry extract combined with maximum antioxidant activity. Therefore, it is necessary to select the optimal-time extraction method and the appropriate solvent for each herb.

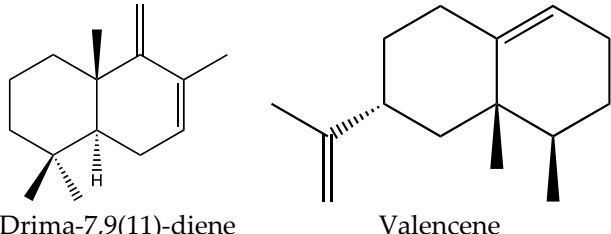

Drima-7,9(11)-diene          Valencene

**Figure 3.** The main compounds identified in essential oils from *Turnera diffusa* extracted by ecofriendly methods.

### 4. Other Ecofriendly Extraction Methods Used to Extract Essential Oils and Antioxidants

In recent years, other methodologies have been used to recover essential oils and phenolic compounds. Supercritical-fluid-assisted extraction, primarily using supercritical

carbon dioxide (SC-CO$_2$), can be used to extract oils from natural products, as it does not produce substantial thermal degradation or contamination by organic solvents. SC-CO$_2$ is a widely used solvent due to its various advantages, such as being cheap, readily available, nontoxic, environmentally friendly and recyclable, and has lower critical points and high diffusivity [110]. Due to the quality of the oils recovered by this technology, as well as all the benefits it represents, various studies have been carried out on the extraction of oils from oregano and laurel, as well as on a large number of aromatic species. Ohmic-assisted extraction is a technology that has a wide range of applications, such as blanching [111,112], evaporation [113,114], dehydration [115], extraction [116,117], and sterilization [118]. It is based on the fact that an alternating current can pass through the sample. Therefore, this technology can heat up samples quickly in short periods. Heat generation occurs within the medium due to its inherent resistance [119]. Electroporation of cell membranes can be induced by CO, and it is assumed that it dominates in experiments performed when electric current passes through biological tissue, causing an increase in temperature and damage to the membrane, resulting in the diffusion of solutes within the cell structure [120,121]. One of the most recent techniques for the extraction of bioactive compounds is accelerated solvent extraction (ASE). Introduced in 1995 as an alternative to the Soxhlet extraction method [122,123], it consists of a stainless-steel cell with electronic control of pressure, temperature, time, and solvent volume. Pressure is applied to the cell (0.2–20 MPa) at a temperature limit of 200 °C by pumping the solvent. Multiple extraction cycles can be used. The process is based on the efficiency of the extraction to break the tertiary structure of the sample proteins and the binding capacity of the sample lipids with respect to the matrix sites. In addition, it is necessary to consider that the increase in temperature and the efficiency of the solvent contact with the sample shorten the times to extract the compounds [124].

These three technologies have been used to extract essential oils and phenolic compounds from sources such as *Geoffroea decorticans* [125], carrot seed [126], *Origanum vulgare* [127], *Laurus nobilis* [128], black rice bran [129], cereal [124], microalgae [123], coffee [130], *Capsicum annum* L. [131], tea leaves [130,132,133], and others. However, there are no reports on the use of these technologies with *Lippia turbinata* and *Turnera diffusa*. Therefore, they represent an opportunity area for their study and the characterization of the compounds obtained. Moreover, the abundance of these plants is wide, and they are already used in traditional medicine and represent a viable alternative for the recovery of compounds using ecofriendly technologies that reduce processing times, eliminate or reduce the use of organic solvents, preserve the integrity of the compounds, and have broader applications with scientific and technological support.

## 5. Preconcentration and Purification of Essential Oils

After using any type of essential oil extraction process, it is necessary to preconcentrate the sample; that is, eliminate the residual water that the sample may contain and separate the oily phase using CH$_2$CH$_2$ [57] or Na$_2$SO$_4$ [63,134–138]. Likewise, to isolate and purify essential oils, high-speed counter-current chromatography (HSCCC) is an excellent and novel semipreparative scale technique [139] used on several traditional herbs and others natural products such as *Alpinia oxyphylla* Miquel [134], *Cuminum cyminum* [139], *Curcuma wenyujin* [140], *Piper claussenianum* [135], *Eucommia ulmoides* Oliv. [141], and *Crataegus pinnatifida* [142]. It is a support-free liquid–liquid partition chromatography that, unlike a conventional column chromatography, eliminates the irreversible adsorption of the sample on the solid support. Since it does not employ This method, it allows a larger injection of relatively pure sample, a short separation time, a high purity of fractions, the use of different two-phase solvents, and an easy scale-up [134]. The solvents most commonly used for essential oils are n-hexane-methanol-water (5:4:1, *v/v*) [134,139], petroleum ether-ethanol-diethyl ether-water (5:4:0.5:1, *v/v*) [140], hexane/acetonitrile (1:1), hexane/methanol (1:1), hexane/acetonitrile/ethyl acetate (1:1:0.4), and hexane/acetonitrile/methanol (1:1:0.5) [135]. To isolate antioxidants, ethyl acetate-ethanolwater (4:1:5, *v/v*), petroleum ether-ethyl

acetate-methanol-water (1:5:1:5, *v/v*), and ethyl acetate-n-butanolwater (1:2:3, *v/v*) [141] are used. It is worth mentioning that this technique has not been used for *Turnera diffusa* and *Lippia turbinata*; therefore, they represent a reliable alternative to isolate and purify the compounds present in essential oils.

## 6. Concluding Remarks

The importance of Lippia turbinata and Turnera diffusa is not only due to their applications in traditional medicine, but also to the fact that the compounds present in these two species are used in the pharmaceutical, cosmetic, and food industries. The use of ecofriendly technologies for the recovery process of bioactive compounds (oils and phenolic compounds) are a viable alternative from the technical–economic point of view, since they allow the acceleration of the extraction process, reduce costs, do not degrade the compounds or leave residues of toxic solvents, and increase yields, thereby reducing the environmental impact. However, it is necessary to design tailor-made processes for each species and choose the best option. All this is because they are two species of great abundance in Mexico, but are little explored. Likewise, the National Forestry Commission (CONAFOR) focuses its efforts on high-impact projects for the development of ecofriendly technologies for the recovery of compounds of high industrial interest from nontimber forestry for the benefit of products with ecofriendly processes.

**Supplementary Materials:** The following are available online at https://www.mdpi.com/article/10.3390/separations8090158/s1, Figure S1: The most recent compounds identified in polyphenols from *Lippia turbinata*.

**Author Contributions:** Conceptualization, writing—original draft preparation, project administration, funding acquisition, R.R.; investigation, writing—review and editing, G.C.G.M.-Á.; writing—review and editing, supervision, P.A.-Z. All authors have read and agreed to the published version of the manuscript.

**Funding:** This research was funded by the National Forestry Commission and the Mexican Council for Science and Technology (CONAFOR–CONACYT) through the Sectoral Fund for Forestry Research, Development, and Technological Innovation within the projects "Estandarización de proceso de extracción de aceites esenciales de especies aromáticas: diseño y construcción de equipo" (number B-S-65769) and "Diseño y construcción de equipo semiautomático para la extracción de cera de candelilla orgánica" (number B-S-131466).

**Data Availability Statement:** Not applicable.

**Acknowledgments:** The authors thank the staff of the National Forestry Commission for the facilities used to develop this project.

**Conflicts of Interest:** The authors declare no conflict of interest.

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
