# Peer review of "Currently Applied Extraction Processes for Secondary Metabolites from Lippia turbinata and Turnera diffusa and Future Perspectives"

_separations, doi:10.3390/separations8090158_

Round 1
Reviewer 1 Report
In the manuscript entitled " Currently applied extraction processes of secondary metabolites from Lippia turbinata and Turnera diffusa and future perspectives" authored by Guillermo Cristian Guadalupe Martínez-Ávila , Pedro Aguilar-Zarate , Romeo Rojas, authors described conventional and alternative extraction methods to be applied in case of two species poleo (Lippia turbinata) and damiana (Turnera diffusa Wild).
I found the article well written and documented, the importance of the topic is clearly stated by the authors. Some modifications should be considered, and the manuscript can be considered for publication after appropriate minor revisions, as detailed in the reviewer’s comments to the Authors
Line 12 add Griseb. (Lippia turbinata Griseb.)
Lines 29 - 30 rephrase to avoid repetition of the words essential oils
Line 32 replace good with a synonym (eg beneficial, valuable)
Line 46 rephrase to avoid repetition of the word bioactive
Line 50 replace properties of inhibition of tyrosinase with tyrosinase inhibitory properties
Lines 55 - 56 replace Is widely used in folk medicine due to its properties as a digestive and antispasmodic to treat gastrointestinal disorder like so food industry for its flavour
with
It is widely used in folk medicine to treat gastrointestinal disorder due to its digestive and antispasmodic properties and in food industry for its flavor.
Line 64 what different sources ? I recommend the authors to develop and to add the corresponding references
Lines 66-67 Therefore, these methods have been accepted as viable alternatives.
I recommend the authors to develop this sentence and to add corresponding references (to point out if these methods are globally/regionally accepted and if there are disadvantages of their use).
Line 112 It is
Line 120 have been reported
Line 122 by conventional methods
Lines 125 -126 consider rephrasing to avoid repetition of is used
Lines 127 – 128 consider synonym to avoid repetition of word reported
Lines 129 – 130 consider rephrasing the sentence (eg. Lippia turbinata belongs to the genus Lippia, thus considered an aromatic species, little documented by the literature)
Lines 135-137 consider rephrasing the sentence and deleting the word however
However, it represents a viable alternative to reduce the conventional chemical additives in food formulation and other areas.
Eg. Given the rich and complex chemical composition, it represents…
Lines 222-225 Consider rephrasing the sentence to avoid repetition of to eliminate or reduce
Lines 227 and 252 by ecofriendly methods.
Author Response
Reviewer 1
In the manuscript entitled " Currently applied extraction processes of secondary metabolites from Lippia turbinata and Turnera diffusa and future perspectives" authored by Guillermo Cristian Guadalupe Martínez-Ávila , Pedro Aguilar-Zarate , Romeo Rojas, authors described conventional and alternative extraction methods to be applied in case of two species poleo (Lippia turbinata) and damiana (Turnera diffusa Wild).
I found the article well written and documented, the importance of the topic is clearly stated by the authors. Some modifications should be considered, and the manuscript can be considered for publication after appropriate minor revisions, as detailed in the reviewer’s comments to the Authors
Point 1. Line 12 add Griseb. (Lippia turbinata Griseb.)
Response 1. Done
Point 2. Lines 29 - 30 rephrase to avoid repetition of the words essential oils
Response 2. Done
Point 3. Line 32 replace good with a synonym (eg beneficial, valuable)
Response 3. Done
Point 4. Line 46 rephrase to avoid repetition of the word bioactive
Response 4. Done
Point 5. Line 50 replace properties of inhibition of tyrosinase with tyrosinase inhibitory properties
Response 5. Done
Point 6. Lines 55 - 56 replace Is widely used in folk medicine due to its properties as a digestive and antispasmodic to treat gastrointestinal disorder like so food industry for its flavour
with
It is widely used in folk medicine to treat gastrointestinal disorder due to its digestive and antispasmodic properties and in food industry for its flavor.
Response 6. Done
Point 7. Line 64 what different sources ? I recommend the authors to develop and to add the corresponding references
Response 7. Done
Point 8. Lines 66-67 Therefore, these methods have been accepted as viable alternatives.
I recommend the authors to develop this sentence and to add corresponding references (to point out if these methods are globally/regionally accepted and if there are disadvantages of their use).
Response 8. The sentence was developed and references were added
Point 9. Line 112 It is
Response 9. Done
Point 10. Line 120 have been reported
Response 10. Done
Point 11. Line 122 by conventional methods
Response 11. Done
Point 12. Lines 125 -126 consider rephrasing to avoid repetition of is used
Response 12. Done
Point 13. Lines 127 – 128 consider synonym to avoid repetition of word reported
Response 13. Done
Point 14. Lines 129 – 130 consider rephrasing the sentence (eg. Lippia turbinata belongs to the genus Lippia, thus considered an aromatic species, little documented by the literature)
Response 14. Done
Point 15. Lines 135-137 consider rephrasing the sentence and deleting the word however
However, it represents a viable alternative to reduce the conventional chemical additives in food formulation and other areas.
Eg. Given the rich and complex chemical composition, it represents…
Response 15. Done
Point 16. Lines 222-225 Consider rephrasing the sentence to avoid repetition of to eliminate or reduce
Response 16. Done
Point 17. Lines 227 and 252 by ecofriendly methods.
Response 17. Done

Reviewer 2 Report
The paper entitled ‘Currently applied extraction processes of secondary metabolites from Lippia turbinata and Turnera diffusa and future perspectives’ presents comprehensive review based on two main species in the Mexico semi-desert. Authors decided to focus on comparison of traditional and eco-friendly extraction methods which were used for obtain essential oil/secondary plant metabolites. On the whole, the review presents interesting and important information nevertheless the paper needs to be improved. Some sections are not readable and information should be enriched. Below I present my suggestions: 1. Keywords: should include ‘monoterpenes’ due to the compounds are main secondary metabolites of the plant 2. Section 2. Traditional extraction methods should be divided into subsections with regard to methods or conditions. In present form the section is hard to read. 3. Table 1. and Table 2. There are little information (extractions methods) . Is it possible to enrich the tables?Author Response
Reviewer 2
The paper entitled ‘Currently applied extraction processes of secondary metabolites from Lippia turbinata and Turnera diffusa and future perspectives’ presents comprehensive review based on two main species in the Mexico semi-desert. Authors decided to focus on comparison of traditional and eco-friendly extraction methods which were used for obtain essential oil/secondary plant metabolites. On the whole, the review presents interesting and important information nevertheless the paper needs to be improved. Some sections are not readable and information should be enriched. Below I present my suggestions:
Point 1. Keywords: should include ‘monoterpenes’ due to the compounds are main secondary metabolites of the plant
Response 1. Done
Point 2. Section 2. Traditional extraction methods should be divided into subsections with regard to methods or conditions. In present form the section is hard to read.
Response 2. Done. Was divided in subsections
Point 3. Table 1. and Table 2. There are little information (extractions methods) . Is it possible to enrich the tables?
Response 3. A single table was made and more information was included

Reviewer 3 Report
The authors presented a review about currently applied extraction processes of secondary metabolites from Lippia turbinata and Turnera diffusa and future perspectives, with accent on eco friendly techniques. I like the idea of the article, However I do not consider that the paper is ready for publication in the current form, and this is why I will suggest a major revision.
Although the extraction techniques were described, some of them, and especially the newest once were omitted. Please include them in the discussion (you will see my comments below). Moreover, the authors did not touch the part of purification and pre-concentration. I suggest them to describe it also in a new chapter, putting the accent on separation between the components. However, other specific comments are presented below.
Line 33 “and represent an alternative in replacing synthetic chemicals” the statement is to wide and somehow inappropriate. One may understand that essential oils can replace any synthetic chemicals. Please rephrase and keep the sentences more focused.
Considering the title I would start the introduction with a short presentation of Lippia turbinata and Turnera diffusa first, and then speaking about the bioactive compounds.
Please point out the originality of the article at the end of introduction.
Please check in the whole text the references style. They are not consistent.
Line 88 – please write it as Table 1
Line 125 – Table 2
Table 1 and 2 can be merged in a single one, by having a blank line where the plant can be written
|
Turnera diffusa |
|||||||||
|
|
|
|
|
|
|
|
|
|
|
|
Lippia turbinata |
|||||||||
|
|
|
|
|
|
|
|
|
|
|
Line 150. Regarding limonene, please mention other sources, and origins. The most important is maybe that it was reported as key exogenous biomarker denoting a deficient liver metabolism (https://dx.doi.org/10.3390/jcm10010032)
Probably the compounds reported in Figure 3 can be moved in supplementary materials. They are well known structures that can be found very easy, and they are not bringing key information to be kept in the main text.
Accelerated solvent extraction (ASE) is a new, modern and eco friendly technique widely used for extraction of bioactive compounds. Please include it in Chapter 3 or Chapter 4 (DOI: 10.1007/s12161-019-01481-z, DOI: 10.1002/elps.201700419). Supercritical fluid extraction was not discussed also, and this is another modern and eco-friendly technique. Please also discuss this in the review (suggested reference DOI: 10.1002/jssc.201801269).
Line 196 – Table 3
Author Response
Reviewer 3
The authors presented a review about currently applied extraction processes of secondary metabolites from Lippia turbinata and Turnera diffusa and future perspectives, with accent on eco friendly techniques. I like the idea of the article, However I do not consider that the paper is ready for publication in the current form, and this is why I will suggest a major revision.
Although the extraction techniques were described, some of them, and especially the newest once were omitted. Please include them in the discussion (you will see my comments below). Moreover, the authors did not touch the part of purification and pre-concentration. I suggest them to describe it also in a new chapter, putting the accent on separation between the components. However, other specific comments are presented below.
Response. The additional information was included into new section.
Point 1. Line 33 “and represent an alternative in replacing synthetic chemicals” the statement is to wide and somehow inappropriate. One may understand that essential oils can replace any synthetic chemicals. Please rephrase and keep the sentences more focused.
Response 1. Done
Point 2. Considering the title I would start the introduction with a short presentation of Lippia turbinata and Turnera diffusa first, and then speaking about the bioactive compounds.
Response 2. Done
Point 3. Please point out the originality of the article at the end of introduction.
Response 3. Done
Point 4. Please check in the whole text the references style. They are not consistent.
Response 4. Done
Point 5. Line 88 – please write it as Table 1
Response 5. Done
Point 6. Line 125 – Table 2
Response 6. Done
Point 7. Table 1 and 2 can be merged in a single one, by having a blank line where the plant can be written
|
Turnera diffusa |
|||||||||
|
|
|
|
|
|
|
|
|
|
|
|
Lippia turbinata |
|||||||||
|
|
|
|
|
|
|
|
|
|
|
Response 7. Done
Point 8. Line 150. Regarding limonene, please mention other sources, and origins. The most important is maybe that it was reported as key exogenous biomarker denoting a deficient liver metabolism (https://dx.doi.org/10.3390/jcm10010032)
Response 8. Done
Point 9. Probably the compounds reported in Figure 3 can be moved in supplementary materials. They are well known structures that can be found very easy, and they are not bringing key information to be kept in the main text.
Response 9. Done. The figure was moved to supplementary material
Point 10. Accelerated solvent extraction (ASE) is a new, modern and eco friendly technique widely used for extraction of bioactive compounds. Please include it in Chapter 3 or Chapter 4 (DOI: 10.1007/s12161-019-01481-z, DOI: 10.1002/elps.201700419). Supercritical fluid extraction was not discussed also, and this is another modern and eco-friendly technique. Please also discuss this in the review (suggested reference DOI: 10.1002/jssc.201801269).
Response 10. Done. Information included in section 4
Point 11. Line 196 – Table 3
Response 11. Done

Round 2
Reviewer 2 Report
Dear Authors,
Thank you for manuscript improvement/correction regarding to my suggestions. Present form of manuscript is more readable and informative. All suggestions were applied. My recommendation: accept in present form.
Reviewer 3 Report
The authors addressed all the comments. I suggest that the manuscript will be accepted in the present form.